# DISTRIBUTION-INTERPOLATION TRADE OFF IN GENERATIVE MODELS

**Damian Leśniak**[*]
Jagiellonian University

**Igor Sieradzki**[*]
Jagiellonian University

**Igor Podolak**
Jagiellonian University

## ABSTRACT

We investigate the properties of multidimensional probability distributions in the context of latent space prior distributions of implicit generative models. Our work revolves around the phenomena arising while decoding linear interpolations between two random latent vectors – regions of latent space in close proximity to the origin of the space are oversampled, which restricts the usability of linear interpolations as a tool to analyse the latent space. We show that the distribution mismatch can be eliminated completely by a proper choice of the latent probability distribution or using non-linear interpolations. We prove that there is a trade off between the interpolation being linear, and the latent distribution having even the most basic properties required for stable training, such as finite mean. We use the multidimensional Cauchy distribution as an example of the prior distribution, and also provide a general method of creating non-linear interpolations, that is easily applicable to a large family of commonly used latent distributions.

## 1 INTRODUCTION

Generative latent variable models have grown to be a very popular research topic, with Variational Auto-Encoders (VAEs) (Kingma & Welling, 2013) and Generative Adversarial Networks (GANs) (Goodfellow et al., 2014) gaining a lot of interest in the last few years. VAEs use a stochastic *encoder* network to embed input data in a typically lower dimensional space, using a conditional probability distribution $p(\mathbf{z}|x)$ over possible latent space codes $z \in \mathbb{R}^D$. A stochastic *decoder* network is then used to reconstruct the original sample. GANs, on the other hand, use a *generator* network that creates data samples from noise $z \sim p(\mathbf{z})$, where $p(\mathbf{z})$ is a fixed prior distribution, and train a *discriminator* network jointly to distinguish between real and generated data.

Both of these model families require a probability distribution to be defined on the latent space. The most popular variants are the multidimensional normal distribution and the uniform distribution on the zero-centred hypercube. Given a trained model, studying the structure of the latent space is a common way to measure generator capabilities.

### 1.1 MOTIVATION BEHIND INTERPOLATIONS

There are various methods used to analyse the latent space. Locally, one can sample and decode points in close neighbourhood of a given latent vector to investigate a small region in the space. On the other hand, global methods are designed to capture long-distance relationships between points in the space, e.g. latent arithmetics, latent directions analysis, and interpolations (see e.g. Mikolov et al. (2013); Kilcher et al. (2017); Radford et al. (2015); White (2016); Agustsson et al. (2017)).

The main advantage of using interpolations is the interpretability that comes with dealing with one-dimensional curves, instead of high-dimensional Euclidean space. For example, if the model has managed to find a meaningful representation, one would expect the latent space to be organised in a way that reflects the internal structure of the training dataset. In that case, decoding an interpolation will show a gradual transformation of one endpoint into the other. Contrarily, if the model memorises the data, the latent space might consist of regions corresponding to particular training examples, divided by boundaries with unnatural, abrupt changes in generated data (Arvanitidis et al., 2017). We

---

[*]These two authors contributed equally
This work was supported by National Science Centre, Poland (grants no. 2015/19/B/ST6/01819).

need to note that this notion of "meaningful representation" is not enforced by the training objective. However, it is not contradicting the objective, making it necessary to use additional tools to evaluate whether the learned manifold is coherently structured and equipped with desirable qualities.

What distinguishes interpolations from other low-dimensional methods is the *shortest path property*. In absence of any additional knowledge about the latent space, it feels natural to use the Euclidean metric. In that case, the shortest path between two points is defined as a segment. This is, probably the most popular, *linear interpolation*, formally defined as $f^L(x_1, x_2, \lambda) = (1-\lambda)x_1 + \lambda x_2$, for $\lambda \in [0, 1]$, where $x_1, x_2$ are the endpoints. Other definitions of shortest path might yield different interpolations, we will study some of them later on.

While traversing the latent space along the shortest path between two points, a well-trained model should transform the samples in a sensible way. For example, if the modelled data has a natural hierarchy, we would expect the interpolation to reflect it, i.e. an image of a truck should not arise on a path between images of a cat and a dog. Also, if the data can be described with a set of features, then an interpolation should maintain any features shared by the endpoints along the path. For example, consider a dataset of images of human faces, with features such as wearing sunglasses, having a long beard, etc. Again, this is not enforced by the training objective. If one would desire such property, it is necessary to somehow include the information about the trained manifold in the interpolation scheme.

There has been an amount of work done on equipping the latent space with a stochastic Riemannian metric (Arvanitidis et al., 2017) that additionally depends on the generator function. The role of the shortest paths is fulfilled by the geodesics, and the metric is defined precisely to enforce some of the properties mentioned above. This approach is somewhat complementary to the one we are concerned with – instead of analysing the latent space using simple tools, we would need to find a more sophisticated metric that describes the latent space comprehensively, and then analyse the metric itself.

If our goal was solely the quality of generated interpolation samples, the aforementioned approach would be preferable. However, in this work we are concerned with evaluating the properties directly connected with the model's objective. With that in mind, we criticise the broad use of linear interpolations in this particular context. In this work we shall theoretically prove that linear interpolations are an incorrect tool for the stated task, and propose a simple, suitable interpolation variant.

### 1.2 THE DISTRIBUTION MISMATCH

While considered useful, the linear interpolation used in conjunction with the most popular latent distributions results in a *distribution mismatch* (also defined in Agustsson et al. (2017); Kilcher et al. (2017)). That is, if we fix the $\lambda$ coefficient and interpolate linearly between two endpoints sampled from the latent space distribution, the probability distribution of the resulting vectors will differ significantly from the latent distribution. This can be partially explained by the well-known fact that in high dimensions the norms of vectors drawn from the latent distribution are concentrated around a certain value. As a consequence, the midpoints of sampled pairs of latent vectors will have, on average, significantly smaller norm. Thus, the linear interpolation oversamples regions in close proximity of the origin of the latent space. A thorough analysis of this phenomenon will be conducted in section 2.1.

Such behaviour raises questions about the applicability of the linear interpolation to study the latent space. Indeed, changing the latent distribution after the model was trained may have unexpected consequences. In Kilcher et al. (2017), experiments conducted using a DCGAN model (Radford et al., 2015) on the celebA dataset (Liu et al., 2015) showed flawed data generation near the latent space origin. Other works concerning the traversal of latent space do not mention this effect, e.g. Agustsson et al. (2017). We recreated this experiment, and concluded that it might be caused by stopping the training process too early (see Appendix C figure 6 for a visualisation). This may explain the apparent disagreement in the literature. Nevertheless, with either a midpoint decoding to a *median face*, or a non-sensible sample, the interpolation is not informative – we would like to see smooth change of features, and not a transition through the same, homogeneous region.

The solution is, either, to change the latent distribution so that the linear interpolation will not cause a distribution mismatch, or redefine the shortest path property. A simple well-known compromise is

to use *spherical interpolations* (Shoemake, 1985; White, 2016). As the latent distribution is concentrated around a sphere, replacing segments with arcs causes relatively small distribution mismatch (see section 3.2). Nonetheless, reducing the consequences of the distribution mismatch is still a popular research topic (Agustsson et al., 2017; Kilcher et al., 2017; Arvanitidis et al., 2017).

### 1.3 MAIN CONTRIBUTIONS

In section 2.1 we show that if the linear interpolation does not change the latent probability distribution, then it must be trivial or "pathological" (with undefined expected value). Then, in section 2.2, we give an example of such an invariant distribution, namely the Cauchy distribution, thus proving its existence. We also discuss the negative consequences of choosing a heavy-tailed probability distribution as the latent prior.

In section 3 we relax the Euclidean shortest path property of interpolations, and investigate non-linear interpolations that do not cause the latent distribution mismatch. We describe a general framework for creating such interpolations, and give two concrete examples in sections 3.4 and 3.5. We find these interpolations to be appropriate for evaluating the model's objective induced properties in contrast to the linear interpolations.

The experiments conducted using the DCGAN model on the CelebA dataset are presented solely to illustrate the problem, not to study the DCGAN itself, theoretically or empirically.

## 2 LATENT DISTRIBUTIONS

In this section we will tackle the problem of distribution mismatch by selecting a proper latent distribution. Let us assume that we want to train a generative model which has a $D$-dimensional latent space and a fixed latent probability distribution, defined by a random variable $\mathbf{Z}$. We denote by $X \sim \mathcal{X}$ that the random variable $X$ has distribution $\mathcal{X}$. $X_n \simeq \mathcal{X}$ represents the fact that the sequence of random variables $\{X_n\}_{n\in\mathbb{N}}$ converges weakly to a random variable with distribution $\mathcal{X}$ as $n$ tends to infinity. By $X_n \simeq \mathcal{X}_n$ we mean that $\lim_{n\to\infty} \sup_{x\in\mathbb{R}} |CDF_{X_n}(x) - CDF_{\mathcal{X}_n}(x)| = 0$, where $CDF_X$ denotes the cumulative distribution function of $X$. The index $n$ will usually be omitted for readability. In other words, by $X \simeq \mathcal{X}$ we mean, informally, that $X$ has distribution similar to $\mathcal{X}$.

### 2.1 LINEAR INTERPOLATION INVARIANCE PROPERTY

**Property 2.1** (**Linear Interpolation Invariance**). *If $\mathbf{Z}$ defines a distribution on the $D$-dimensional latent space, $\mathbf{Z}^{(1)}$ and $\mathbf{Z}^{(2)}$ are independent and distributed identically to $\mathbf{Z}$, and for every $\lambda \in [0, 1]$ the random variable $f^L(\mathbf{Z}^{(1)}, \mathbf{Z}^{(2)}, \lambda) := (1 - \lambda)\mathbf{Z}^{(1)} + \lambda\mathbf{Z}^{(2)}$ is distributed identically to $\mathbf{Z}$, then we will say that $\mathbf{Z}$ has the linear interpolation invariance property, or that linear interpolation does not change the distribution of $\mathbf{Z}$.*

The most commonly used latent probability distributions $\mathbf{Z}$ are products of $D$ independent random variables. That is, $\mathbf{Z} = (Z_1, Z_2, \ldots, Z_D)$, where $Z_1, Z_2, \ldots, Z_D$ are the independent marginals distributed identically to $Z$. If the norms of $\mathbf{Z}$ concentrate around a certain value, then the latent distribution resembles sampling from a zero-centred sphere and the linear interpolation oversamples regions in the proximity of the origin of the latent space. As a consequence, $\mathbf{Z}$ does not have the linear interpolation invariance property. The following observation will shed light upon this problem. Let $\mathcal{N}(\mu, \sigma^2)$ denote the normal distribution with mean $\mu$ and variance $\sigma^2$.

**Observation 2.1.** *Let us assume that $Z^2$ has finite mean $\mu$ and finite variance $\sigma^2$. If $\mu > 0$, then $\|\mathbf{Z}\| \simeq \mathcal{N}\big(\sqrt{D\mu}, \frac{\sigma^2}{4\mu}\big)$ as $D \to \infty$. If $\mu = 0$, then $\|\mathbf{Z}\| = 0$ almost everywhere.*

The proof of this and all further observations is presented in the appendix B.

For example, if $Z \sim \mathcal{N}(0, 1)$, then $\mathbf{Z}$ is distributed according to the $D$-dimensional normal distribution with mean $\mathbf{0}$ and identity covariance matrix $\mathbf{I}$. $Z^2$ has moments $\mu = 1$, $\sigma^2 = 2$, thus $\|\mathbf{Z}\| \simeq \mathcal{N}\big(\sqrt{D}, \frac{1}{2}\big)$. The second example is $Z \sim \mathcal{U}(-1, 1)$, where $\mathcal{U}(a, b)$ is the uniform distribu-

---

If $Z \sim \mathcal{N}(0, 1)$, then $\|\mathbf{Z}\|$ is distributed according to the chi distribution, equal to the square root of the chi-squared distribution.

tion on the interval $[a, b]$, and $\mathbf{Z}$ is distributed uniformly on the hypercube $[-1, 1]^D$. In that case, $Z^2$ has moments $\mu = \frac{1}{3}$, $\sigma^2 = \frac{4}{45}$, thus $\|\mathbf{Z}\| \simeq \mathcal{N}\left(\sqrt{\frac{D}{3}}, \frac{1}{15}\right)$.

It is worth noting that the variance of the approximated probability distribution of $\|\mathbf{Z}\|$, the *thickness* of the sphere, does not change as $D$ tends to infinity – only the radius of the sphere is affected. On the other hand, if the latent distribution is normalised (divided by the expected value of $\|\mathbf{Z}\|$), then the distribution concentrates around the unit sphere (not necessarily uniformly), and we observe the so-called *soap bubble* phenomenon (Ferenc, 2017).

One might think that the factorisation of the latent probability distribution is the main reason why the linear interpolation changes the distribution. Unfortunately, this is not the case. Let $\overline{\mathbf{Z}} := \frac{1}{2}(\mathbf{Z}^{(1)} + \mathbf{Z}^{(2)})$, where $\mathbf{Z}^{(1)}, \mathbf{Z}^{(2)}$ are two independent samples from $\mathbf{Z}$. Therefore, $\overline{\mathbf{Z}}$ is the distribution of the middle points of a linear interpolation between two vectors drawn independently from $\mathbf{Z}$.

**Observation 2.2.** *If $\mathbf{Z}$ has a finite mean, and $\overline{\mathbf{Z}}$ is distributed identically to $\mathbf{Z}$, then $\mathbf{Z}$ must be concentrated at a single point.*

If a probability distribution is not heavy-tailed, then its tails are bounded by the exponential distribution, which in turn means that it has a finite mean. Therefore, all distributions having undefined expected value must be heavy-tailed. We will refer to this later on, as the heavy tails may have strong negative impact on the training procedure.

There have been attempts to find $\mathbf{Z}$, with finite mean, such that $\overline{\mathbf{Z}}$ is at least similar to $\mathbf{Z}$. Kilcher et al. (2017) managed to reduce the distribution mismatch by defining the latent distribution as

$$\mathbf{V} \sim \mathcal{U}(\mathcal{S}^{D-1}), \quad r \sim \Gamma(\frac{1}{2}, \theta), \quad \theta > 0, \quad \mathbf{Z} = \sqrt{r}\mathbf{V},$$

where $\mathcal{U}(\mathcal{S}^{D-1})$ is the uniform distribution on the unit sphere, and $\Gamma(\frac{1}{2}, \theta)$ is the gamma distribution. We extend this idea by using a distribution that has no finite mean, namely the Cauchy distribution.

## 2.2 THE CAUCHY DISTRIBUTION

The standard Cauchy distribution is denoted by $\mathcal{C}(0, 1)$, and its density function is defined as $1/\left(\pi(1 + x^2)\right)$. The most important property of the Cauchy distribution is the fact that if $C^{(1)}, \ldots, C^{(n)}$ are independent samples from the standard Cauchy distribution, and $\lambda_1, \ldots, \lambda_n \in [0, 1]$ with $\lambda_1 + \ldots + \lambda_n = 1$, then $\lambda_1 C^{(1)} + \ldots + \lambda_n C^{(n)}$ is also distributed according to the standard Cauchy distribution. In case of $n = 2$ it means that the Cauchy distribution satisfies the distribution matching property. On the other hand, as a consequence of observation 2.2, the Cauchy distribution cannot have finite mean. In fact, all of its moments of order greater than or equal to one are undefined. See Siegrist (2017) for further details.

There are two ways of using the Cauchy distribution in high dimensional spaces while retaining the distribution matching property. The *multidimensional* Cauchy distribution is defined as a product of independent standard Cauchy distributions. Then, the linear interpolation invariance property can be simply proved by applying the above formulas coordinate-wise. In the case of vectors drawn from the multidimensional Cauchy distribution we may expect that some of the coordinates will be sufficiently larger, by absolute value, than the others (Hansen et al., 2006), thus making the latent distribution similar to coordinate-wise sampling.

In contrast, the *multivariate* Cauchy distribution comes with the isotropy property at the cost of the canonical directions becoming statistically dependent. There are multiple ways of defining it, and further analysis is out of the scope of this paper. We tested both variants as latent distributions with similar results. From now on, we shall concentrate on the non-isotropic Cauchy distribution.

The Cauchy distribution is a member of the family of *stable distributions*, and has been previously used to model heavy-tailed data (Nolan, 2018). However, according to our best knowledge, the Cauchy distribution has never been used as the latent distribution in generative models. Figure 1 presents a decoded linear interpolations between random latent vectors using a DCGAN model trained on the CelebA dataset for the Cauchy distribution and the distribution from Kilcher et al. (2017).

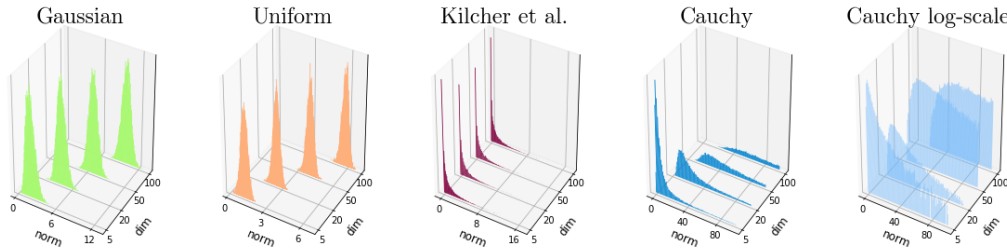

Figure 1: Comparison of linear interpolations from DCGAN trained on the Cauchy distribution (top) and one trained on the distribution proposed by Kilcher et al. (2017) (bottom).

It should be noted that if $D$ is large enough, the distribution of the norms of vectors sampled from the $D$-dimensional Cauchy distribution has a low density near zero – similarly to the normal and uniform distributions – but linear interpolations do not oversample this part of the latent space, due to the heavy-tailed nature of the Cauchy distribution. Comparison of the distributions of norms is given in Figure 2.

Figure 2: Illustration of observation 2.1: distributions of Euclidean norms of latent vectors, for different probability distributions, with increasing latent space dimension.

The *distribution-interpolation trade off* states that if the probability distribution has the linear interpolation invariance property, then it must be trivial or heavy-tailed. In case of the Cauchy distribution we observed issues with generating images if the norm of the sampled latent vector was relatively large (the probability distribution of the norms is also heavy-tailed). Some of those faulty examples are presented in the appendix C. This is consistent with the known fact, that artificial networks perform poorly if their inputs are not normalised (see e.g. Glorot & Bengio (2010)).

A probability distribution having the linear interpolation invariance property cannot be normalised using linear transformations. For example, the batch normalisation technique (Ioffe & Szegedy, 2015) would be highly ineffective, as the mean of a batch of samples is, in fact, a single sample from the distribution. On the other hand, using a non-linear normalisation (e.g., clipping the norm of the latent vectors in subsequent layers), is mostly equivalent to changing the latent probability distribution and making the interpolation non-linear. This idea will be explored in the next section.

## 3    INTERPOLATIONS

In this section we review the most popular variants of interpolations, with an emphasis on the distribution mismatch analysis. We also present two new examples of interpolations stemming from a general scheme, that perform well with the popular latent priors.

An *interpolation* on the latent space $\mathbb{R}^D$ is formally defined as a function

$$f : \mathbb{R}^D \times \mathbb{R}^D \times [0,1] \ni (x_1, x_2, \lambda) \mapsto x \in \mathbb{R}^D.$$

For brevity, we will represent $f(x_1, x_2, \lambda)$ by $f_{x_1, x_2}(\lambda)$.

**Property 3.1 (Distribution Matching Property).** *If $\mathbf{Z}$ defines a distribution on the $D$-dimensional latent space, $\mathbf{Z}^{(1)}$ and $\mathbf{Z}^{(2)}$ are independent and distributed identically to $\mathbf{Z}$, and for every $\lambda \in [0,1]$ the random variable $f_{\mathbf{Z}^{(1)}, \mathbf{Z}^{(2)}}(\lambda)$ is distributed identically to $\mathbf{Z}$, then we will say that the interpolation $f$ has the distribution matching property in conjunction with $\mathbf{Z}$, or that the interpolation $f$ does not change the distribution of $\mathbf{Z}$.*

### 3.1 LINEAR INTERPOLATION

The *linear interpolation* is defined as $f^L_{x_1,x_2}(\lambda) = (1 - \lambda)x_1 + \lambda x_2$. This interpolation does not satisfy the distribution matching property for the most commonly used probability distributions, as they have a finite mean. A notable exception is the Cauchy distribution. This was discussed in details in the previous section.

### 3.2 SPHERICAL LINEAR INTERPOLATION

As in Shoemake (1985); White (2016), the *spherical linear interpolation* is defined as

$$f^{SL}_{x_1,x_2}(\lambda) = \frac{\sin\left[(1 - \lambda)\Omega\right]}{\sin\Omega}x_1 + \frac{\sin[\lambda\Omega]}{\sin\Omega}x_2,$$

where $\Omega$ is the angle between vectors $x_1$ and $x_2$. Note that this interpolation is undefined for parallel endpoint vectors, and the definition cannot be extended without losing the continuity. Also, if vectors $x_1$ and $x_2$ have the same length $R$, then the interpolation corresponds to a geodesic on the sphere of radius $R$. In this regard, it might be said that the spherical linear interpolation is defined as the shortest path on the sphere. The most important fact is that this interpolation can have the distribution matching property.

**Observation 3.1.** *If* **Z** *has uniform distribution on the zero-centred sphere of radius $R > 0$, then $f^{SL}$ does not change the distribution of* **Z***.*

### 3.3 NORMALISED INTERPOLATION

Introduced in Agustsson et al. (2017), the *normalised interpolation* is defined as

$$f^N_{x_1,x_2}(\lambda) = \frac{(1 - \lambda)x_1 + \lambda x_2}{\sqrt{(1 - \lambda)^2 + \lambda^2}}.$$

**Observation 3.2.** *If* $\mathbf{Z} \sim \mathcal{N}(\mathbf{0}, \mathbf{I})$*, then $f^N$ does not change the distribution of* **Z***.*

If vectors $x_1$ and $x_2$ are orthogonal and have equal length, then the curve defined by this interpolation is equal to the one of the spherical linear interpolation. On the other hand, the normalised interpolation behaves poorly if $x_1$ is close to $x_2$. In the extreme case of $x_1 = x_2$ the interpolation is *not* constant with respect to $\lambda$, which violates any sensible definition of the shortest path.

### 3.4 CAUCHY-LINEAR INTERPOLATION

Here we present a general way of designing interpolations that have the distribution matching property in conjunction with a given probability distribution **Z**. This method requires some additional assumptions about **Z**, but it works well with the most popular latent distributions.

Let $L$ be the $D$-dimensional latent space, **Z** define the probability distribution on the latent space, **C** be distributed according to the $D$-dimensional Cauchy distribution on $L$, $K$ be a subset of $L$ such that **Z** is concentrated on this set, and $g : L \to K$ be a bijection such that $g(\mathbf{C})$ is distributed identically to **Z** on $K$. Then for $x_1, x_2 \in K$ we define the *Cauchy-linear interpolation as*

$$f^{CL}_{x_1,x_2}(\lambda) = g\big((1 - \lambda)g^{-1}(x_1) + \lambda g^{-1}(x_2)\big).$$

In other words, for endpoints $x_1, x_2 \sim \mathbf{Z}$:

1. Transform $x_1$ and $x_2$ using $g^{-1}$. This step changes the latent distribution to the $D$-dimensional Cauchy distribution.

2. Linearly interpolate between the transformations to get $x_\lambda = (1 - \lambda)g^{-1}(x_1) + \lambda g^{-1}(x_2)$ for all $\lambda \in [0, 1]$. The transformed latent distribution remains unchanged.

---

Originally referred to as *distribution matched*.

3. Transform $x_\lambda$ back to the original space using $g$. We end up with the original latent distribution.

**Observation 3.3.** *With the above assumptions about g the Cauchy-linear interpolation does not change the distribution of* **Z**.

Finding an appropriate function $g$ might seem hard, but in practice it usually is fairly straightforward. For example, if **Z** is distributed identically to the product of $D$ independent one-dimensional distributions $Z$, then we can define $g$ as $CDF_C^{-1} \circ CDF_Z$ applied to every coordinate.

### 3.5 SPHERICAL CAUCHY-LINEAR INTERPOLATION

We might want the interpolation to have some other desired properties. For example, to behave exactly as the spherical linear interpolation if only the endpoints have equal norm. For that purpose, we need to make additional assumptions. Let **Z** be isotropic, $C$ be distributed according to the one-dimensional Cauchy distribution, and $g : \mathbb{R} \to (0, +\infty)$ be a bijection such that $g(C)$ is distributed identically as $\|\mathbf{Z}\|$ on $(0, +\infty)$. Then we can modify the spherical linear interpolation formula to define what we call the *spherical Cauchy-linear interpolation*

$$f_{x_1,x_2}^{SCL}(\lambda) = \left( \frac{\sin\left[(1-\lambda)\Omega\right]}{\sin\Omega} \frac{x_1}{\|x_1\|} + \frac{\sin[\lambda\Omega]}{\sin\Omega} \frac{x_2}{\|x_2\|} \right) \left[ g\big( (1-\lambda)g^{-1}(\|x_1\|) + \lambda g^{-1}(\|x_2\|) \big) \right],$$

where $\Omega$ is the angle between vectors $x_1$ and $x_2$. In other words:

1. Interpolate the *directions* of latent vectors using the spherical linear interpolation.
2. Interpolate the *norms* using the Cauchy-linear interpolation.

**Observation 3.4.** *With the above assumptions about g, the spherical Cauchy-linear interpolation does not change the distribution of* **Z** *if the* **Z** *distribution is isotropic.*

The simplest candidate for the $g$ function is $CDF_C^{-1} \circ CDF_{\|\mathbf{Z}\|}$, but we usually need to know more about **Z** to check if the assumptions hold. For example, let **Z** be a $D$-dimensional normal distribution with zero mean and identity covariance matrix. Then $\|\mathbf{Z}\| \sim \sqrt{\chi_D^2}$ and

$$CDF_{\sqrt{\chi_D^2}}(x) = CDF_{\chi_D^2}(x^2) = \frac{1}{\Gamma(D/2)}\gamma\left(\frac{D}{2}, \frac{x^2}{2}\right), \text{ for every } x \geq 0,$$

where $\Gamma$ denotes the gamma function, and $\gamma$ is the lower incomplete gamma function. Thus we set $g(x) = \big(CDF_C^{-1} \circ CDF_{\chi_D^2}\big)(x^2)$, with $g^{-1}(x) = \sqrt{\big(CDF_{\chi_D^2}^{-1} \circ CDF_C\big)(x)}$.

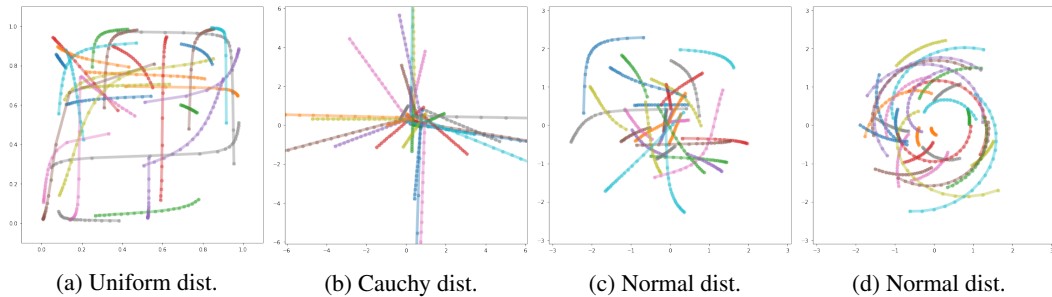

| (a) Uniform dist. | (b) Cauchy dist. | (c) Normal dist. | (d) Normal dist. |

Figure 3: Comparison of the Cauchy-linear (a, b, and c) and the Spherical Cauchy-linear (d) interpolations on a 2D plane for data pairs sampled from different distributions. The Cauchy-linear interpolation in conjunction with the Cauchy distribution naturally results in segments.

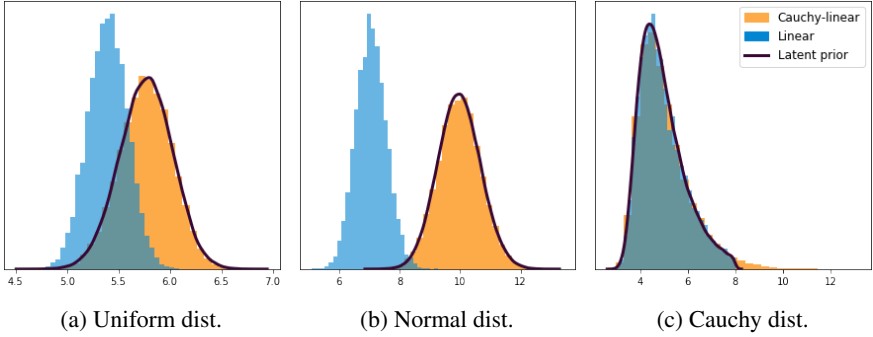

| (a) Uniform dist. | (b) Normal dist. | (c) Cauchy dist. |

Figure 4: Illustration of observation 3.3: comparison of norms of interpolation mid-points for linear (blue), Cauchy-linear (orange) interpolations, and latent prior distribution (dark line) for different latent distributions.

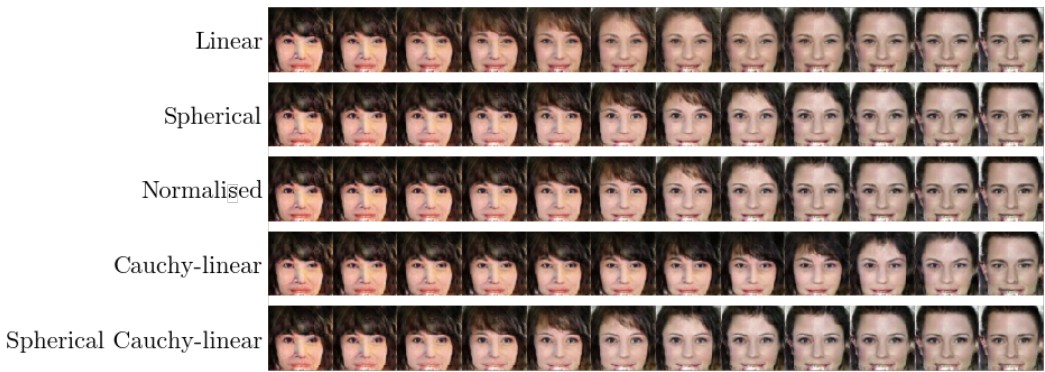

Figure 5: Images generated using a DCGAN model trained on the standard Normal distribution. Comparison of the five interpolations.

Figure 3 shows comparison of the Cauchy-linear and the spherical Cauchy-linear interpolations on a two-dimensional plane for pairs of vectors sampled from different probability distributions. It illustrates how these interpolations manage to keep the distributions unchanged. Figure 4 is an illustration of distribution matching property for Cauchy-linear interpolation. We also compare the data samples generated by the DCGAN model trained on the CelebA dataset; the results are shown in figure 5.

## 4 SUMMARY

We investigated the properties of multidimensional probability distributions in the context of generative models. We found out that there is a certain trade-off: it is impossible to define a latent probability distribution with a finite mean and the linear interpolation invariance property. The $D$-dimensional Cauchy distribution serves as an example of a latent probability distribution that remains unchanged by linear interpolation, at the cost of poor model performance, due to the heavy-tailed nature.

Instead of using the Cauchy distribution as the latent distribution, we propose to use it to define non-linear interpolations that have the distribution matching property. The assumption of the shortest path being a straight line must be relaxed, but our scheme is general enough to provide a way of incorporating other desirable properties.

We observe that there are three different goals when using interpolations for studying a generative model. Firstly, to check whether the training objective was fulfilled, one must use an interpolation that does not cause the distribution mismatch. This is, in our opinion, a necessary step before

performing any further evaluation of the trained model. Secondly, if one is interested in the manifold convexity, linear interpolations are a suitable method provided the above analysis yields positive results. Finally, to perform a complete investigation of the learned manifold one can employ methods that incorporate some information about the trained model, e.g. the approach of Arvanitidis et al. (2017) mentioned in section 1.1.

We do not propose to completely abandon the use of linear interpolations, as the convexity of the learned manifold is still an interesting research topic. For instance, we have observed that generative models are capable of generating sensible images from seemingly out-of-distribution regions, e.g. the emergence of the median face mentioned in the introduction. In our opinion, this is a promising direction for future research.

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

## A   EXPERIMENTAL SETUP

All experiments were conducted using a DCGAN model (Radford et al., 2015), in which the generator network consisted of a linear layer with 8192 neurons, followed by four convolution transposition layers, each using $5 \times 5$ filters and strides of 2, with number of filters in order of layers: 256, 128, 64, 3. Except for the output layer, where *tanh* activation function was used, all previous layers used *ReLU*. Discriminator's architecture mirrored the one from the generator, with a single exception of using *leaky ReLU* instead of vanilla *ReLU* function for all except the last layer. No batch normalisation was used in both networks. Adam optimiser with learning rate of $2e^{-4}$ and momentum set to 0.5 was used. Batch size 64 was used throughout all experiments. If not explicitly stated otherwise, latent space dimension was set to 100. For the CelebA dataset we resized the input images to $64 \times 64$.

## B   PROOFS

**Observation 2.1.** *Let us assume that* $Z^2$ *has finite mean* $\mu$ *and finite variance* $\sigma^2$. *If* $\mu > 0$, *then* $\|\mathbf{Z}\| \simeq \mathcal{N}\big(\sqrt{D\mu}, \frac{\sigma^2}{4\mu}\big)$ *as* $D \to \infty$. *If* $\mu = 0$, *then* $\|\mathbf{Z}\| = 0$ *almost everywhere.*

*Proof.* Recall that $Z, Z_1, \ldots, Z_D$ are independent and identically distributed. Therefore $Z^2, Z_1^2, \ldots, Z_D^2$ are also independent and identically distributed. $\mathbf{Z} = (Z_1, \ldots, Z_D)$ and $\|\mathbf{Z}\|^2 = Z_1^2 + \ldots + Z_D^2$.

$Z^2 \geq 0$, therefore $\mu \geq 0$. If $\mu = 0$, then $Z^2 = 0$ almost everywhere, $\mathbf{Z}^2 = 0$ almost everywhere, $\mathbf{Z} = 0$ almost everywhere, and finally $\|\mathbf{Z}\| = 0$ almost everywhere. From now on we will assume that $\mu > 0$.

Using the central limit theorem we know that $\sqrt{D}\big(\frac{Z_1^2 + \ldots + Z_D^2}{D} - \mu\big)$ converges in distribution to $\mathcal{N}(0, \sigma^2)$ with $D \to \infty$. The convergence of cumulative distribution functions is uniform, because the limit is continuous everywhere

$$\forall_{\epsilon > 0} \exists_{\mathcal{D} > 0} \forall_{D > \mathcal{D}, D \in \mathbb{N}} \forall_{x \in \mathbb{R}} : \left| Pr\Big(\sqrt{D}\Big(\frac{Z_1^2 + \ldots + Z_D^2}{D} - \mu\Big) \leq x\Big) - CDF_{\mathcal{N}(0, \sigma^2)}(x) \right| < \epsilon.$$

$D > 0$, thus

$$Pr\Big(\sqrt{D}\Big(\frac{Z_1^2 + \ldots + Z_D^2}{D} - \mu\Big) \leq x\Big) = Pr\big(Z_1^2 + \ldots + Z_D^2 \leq D\mu + x\sqrt{D}\big)$$
$$= CDF_{\|\mathbf{Z}\|^2}\big(D\mu + x\sqrt{D}\big).$$

Additionally,
$$CDF_{\mathcal{N}(0, \sigma^2)}(x) = CDF_{\mathcal{N}(D\mu, D\sigma^2)}\big(D\mu + x\sqrt{D}\big),$$

and now we have

$$\forall_{\epsilon > 0} \exists_{\mathcal{D} > 0} \forall_{D > \mathcal{D}, D \in \mathbb{N}} \forall_{x \in \mathbb{R}} : \left| CDF_{\|\mathbf{Z}\|^2}\big(D\mu + x\sqrt{D}\big) - CDF_{\mathcal{N}(D\mu, D\sigma^2)}\big(D\mu + x\sqrt{D}\big) \right| < \epsilon.$$

Finally, the function
$$\mathbb{R} \ni x \mapsto D\mu + x\sqrt{D} \in \mathbb{R}$$

is a bijection (again, because $D > 0$), so we may substitute $D\mu + x\sqrt{D}$ with $x$ and the innermost statement will hold for every $x \in \mathbb{R}$

$$\forall_{\epsilon > 0} \exists_{\mathcal{D} > 0} \forall_{D > \mathcal{D}, D \in \mathbb{N}} \forall_{x \in \mathbb{R}} : \left| CDF_{\|\mathbf{Z}\|^2}(x) - CDF_{\mathcal{N}(D\mu, D\sigma^2)}(x) \right| < \epsilon. \tag{1}$$

Before taking square root of the normal distribution we must deal with negative values. Let $\mathcal{N}_+(\nu, \tau)$ be defined by its cumulative distribution function:

$$CDF_{\mathcal{N}_+(\nu, \tau)}(x) = \begin{cases} 0 & \text{if } x < 0, \\ CDF_{\mathcal{N}(\nu, \tau)}(x) & \text{if } x \geq 0. \end{cases}$$

The idea is to take all negative values of $\mathcal{N}(\nu, \tau)$ and concentrate them at zero.

Now we can modify (1)

$$\forall_{\epsilon>0}\exists_{\mathcal{D}>0}\forall_{D>\mathcal{D},D\in\mathbb{N}}\forall_{x\in\mathbb{R}} : \left|CDF_{\|\mathbf{Z}\|^2}(x) - CDF_{\mathcal{N}_+(D\mu,D\sigma^2)}(x)\right| < \epsilon\,, \qquad (2)$$

for $x \geq 0$ we simply use (1), for $x < 0$ the inequality simplifies to $|0 - 0| < \epsilon$.

Since $\|\mathbf{Z}\|^2$ and $\mathcal{N}_+(D\mu, D\sigma^2)$ are non-negative, we are allowed to take the square root of these random variables. The square root is a strictly increasing function, thus for $x \geq 0$ we have

$$CDF_{\mathcal{N}_+(D\mu,D\sigma^2)}(x^2) = CDF_{\sqrt{\mathcal{N}_+(D\mu,D\sigma^2)}}(x) \quad \text{and} \quad CDF_{\|\mathbf{Z}\|^2}(x^2) = CDF_{\|\mathbf{Z}\|}(x)\,,$$

therefore we can approximate the variable $\|\mathbf{Z}\|$

$$\forall_{\epsilon>0}\exists_{\mathcal{D}>0}\forall_{D>\mathcal{D},D\in\mathbb{N}}\forall_{x\in\mathbb{R}} : \left|CDF_{\|\mathbf{Z}\|}(x) - CDF_{\sqrt{\mathcal{N}_+(D\mu,D\sigma^2)}}(x)\right| < \epsilon\,, \qquad (3)$$

for $x \geq 0$ we substitute $x^2$ for $x$ in (2), for $x < 0$ the inequality simplifies, again, to $|0 - 0| < \epsilon$.

This paragraph is a summary of the second part of the proof. To calculate $\sqrt{\mathcal{N}_+(D\mu, D\sigma^2)}$ we observe that, informally, in proximity of $D\mu$ the square root behaves approximately like scaling with constant $(2\sqrt{D\mu})^{-1}$. Additionally, $\mathcal{N}(D\mu, D\sigma^2)$ has *width* proportional to $\sqrt{D}$, which is infinitesimally smaller than $D\mu$, so we expect the result to be

$$\sqrt{\mathcal{N}_+(D\mu, D\sigma^2)} \simeq \mathcal{N}\left(\sqrt{D\mu}, \frac{\sigma^2}{4\mu}\right).$$

Let us define

$$b_\epsilon = \begin{cases} CDF^{-1}_{\mathcal{N}(0,\sigma^2/(4\mu))}(1 - \epsilon) & \text{if } \epsilon \in (0, \frac{1}{2})\,, \\ 0 & \text{if } \epsilon \geq \frac{1}{2}\,. \end{cases}$$

Here $b_\epsilon$ is defined so that the probability of $x$ drawn from $\mathcal{N}\left(\sqrt{D\mu}, \frac{\sigma^2}{4\mu}\right)$ being at least $b_\epsilon$ far from the mean is equal to $2\epsilon$. Also, note that $b_\epsilon$ does not depend on $D$. For now we will assume that $\sqrt{D\mu} - b_\epsilon > 0$ – this is always true for sufficiently large $D$, as $\mu > 0$

$$\forall_{\epsilon>0}\exists_{\mathcal{D}>0}\forall_{D>\mathcal{D},D\in\mathbb{N}} : \sqrt{D\mu} - b_\epsilon > 0\,. \qquad (4)$$

Now let us assume that we have a fixed $\epsilon > 0$. For $x \in [-b_\epsilon, b_\epsilon]$ we write the following inequalities

$$D\mu + 2x\sqrt{D\mu} \leq \left(\sqrt{D\mu} + x\right)^2 \leq D\mu + 2x\sqrt{D\mu} + b_\epsilon^2\,,$$

which are equivalent to $0 \leq x^2 \leq b_\epsilon^2$, thus true.

Every cumulative distribution function is weakly increasing, therefore

$$CDF_{\mathcal{N}(D\mu,D\sigma^2)}\left(D\mu + 2x\sqrt{D\mu}\right) \leq CDF_{\mathcal{N}(D\mu,D\sigma^2)}\left(\left(\sqrt{D\mu} + x\right)^2\right) \leq$$
$$\leq CDF_{\mathcal{N}(D\mu,D\sigma^2)}\left(D\mu + 2x\sqrt{D\mu} + b_\epsilon^2\right).$$

Because we assumed that $\left(\sqrt{D\mu} + x\right)^2 > 0$ for $x \in [-b_\epsilon, b_\epsilon]$, we can replace $\mathcal{N}(D\mu, D\sigma^2)$ with $\mathcal{N}_+(D\mu, D\sigma^2)$

$$CDF_{\mathcal{N}(D\mu,D\sigma^2)}\left(D\mu + 2x\sqrt{D\mu}\right) \leq CDF_{\mathcal{N}_+(D\mu,D\sigma^2)}\left(\left(\sqrt{D\mu} + x\right)^2\right) \leq$$
$$\leq CDF_{\mathcal{N}(D\mu,D\sigma^2)}\left(D\mu + 2x\sqrt{D\mu} + b_\epsilon^2\right).$$

We transform the outer distributions using basic properties of the normal distribution. We also take square root of the middle distribution and obtain

$$CDF_{\mathcal{N}(\sqrt{D\mu},\sigma^2/(4\mu))}\left(\sqrt{D\mu} + x\right) \leq CDF_{\sqrt{\mathcal{N}_+(D\mu,D\sigma^2)}}\left(\sqrt{D\mu} + x\right) \leq$$
$$\leq CDF_{\mathcal{N}(\sqrt{D\mu},\sigma^2/(4\mu))}\left(\sqrt{D\mu} + x + b_\epsilon^2/\left(2\sqrt{D\mu}\right)\right). \qquad (5)$$

$b_\epsilon^2/(2\sqrt{D\mu}) \to 0$ with $D \to \infty$ and $CDF_{\mathcal{N}(\sqrt{D\mu},\sigma^2/(4\mu))}$ is continuous, thus we have uniform convergence

$$\forall_{\epsilon>0}\exists_{\mathcal{D}>0}\forall_{D>\mathcal{D},D\in\mathbb{N}}\forall_{x\in\mathbb{R}}:$$
$$\left| CDF_{\mathcal{N}\left(\sqrt{D\mu},\sigma^2/(4\mu)\right)}\left(\sqrt{D\mu}+x\right) - CDF_{\mathcal{N}(\sqrt{D\mu},\sigma^2/(4\mu))}\left(\sqrt{D\mu}+x+b_\epsilon^2/(2\sqrt{D\mu})\right) \right| < \epsilon.$$

Using (5) we get

$$\forall_{\epsilon>0}\exists_{\mathcal{D}>0}\forall_{D>\mathcal{D},D\in\mathbb{N}}\forall_{x\in[-b_\epsilon,b_\epsilon]}: \left[ \sqrt{D\mu}-b_\epsilon > 0 \implies \right.$$
$$\left. \left| CDF_{\mathcal{N}\left(\sqrt{D\mu},\sigma^2/(4\mu)\right)}\left(\sqrt{D\mu}+x\right) - CDF_{\sqrt{\mathcal{N}_+(D\mu,D\sigma^2)}}\left(\sqrt{D\mu}+x\right) \right| < \epsilon \right]. \quad (6)$$

Now we will extend this result to all $x \in \mathbb{R}$. For $\epsilon > 0$ we have

$$CDF_{\mathcal{N}\left(\sqrt{D\mu},\sigma^2/(4\mu)\right)}\left(\sqrt{D\mu}-b_\epsilon\right) \leq \epsilon, \quad (7)$$

$$CDF_{\mathcal{N}\left(\sqrt{D\mu},\sigma^2/(4\mu)\right)}\left(\sqrt{D\mu}+b_\epsilon\right) \geq 1-\epsilon. \quad (8)$$

Substituting $-b_\epsilon$ and $b_\epsilon$ for $x$ in (6), and using (7) and (8) respectively, we obtain

$$\forall_{\epsilon>0}\exists_{\mathcal{D}>0}\forall_{D>\mathcal{D},D\in\mathbb{N}}:CDF_{\sqrt{\mathcal{N}_+(D\mu,D\sigma^2)}}\left(\sqrt{D\mu}-b_\epsilon\right) < 2\epsilon, \quad (9)$$

$$\forall_{\epsilon>0}\exists_{\mathcal{D}>0}\forall_{D>\mathcal{D},D\in\mathbb{N}}:CDF_{\sqrt{\mathcal{N}_+(D\mu,D\sigma^2)}}\left(\sqrt{D\mu}+b_\epsilon\right) > 1-2\epsilon. \quad (10)$$

Cumulative distribution functions are increasing functions with values in $[0,1]$, thus combining (7) and (9)

$$\forall_{\epsilon>0}\forall_{x<-b_\epsilon}: 0 \leq CDF_{\mathcal{N}\left(\sqrt{D\mu},\sigma^2/(4\mu)\right)}\left(\sqrt{D\mu}+x\right) \leq \epsilon,$$

$$\forall_{\epsilon>0}\exists_{\mathcal{D}>0}\forall_{D>\mathcal{D},D\in\mathbb{N}}\forall_{x<-b_\epsilon}: 0 \leq CDF_{\sqrt{\mathcal{N}_+(D\mu,D\sigma^2)}}\left(\sqrt{D\mu}+x\right) < 2\epsilon,$$

$$\forall_{\epsilon>0}\exists_{\mathcal{D}>0}\forall_{D>\mathcal{D},D\in\mathbb{N}}\forall_{x<-b_\epsilon}:$$
$$\left| CDF_{\mathcal{N}\left(\sqrt{D\mu},\sigma^2/(4\mu)\right)}\left(\sqrt{D\mu}+x\right) - CDF_{\sqrt{\mathcal{N}_+(D\mu,D\sigma^2)}}\left(\sqrt{D\mu}+x\right) \right| < 2\epsilon. \quad (11)$$

Analogically, using (8) and (10)

$$\forall_{\epsilon>0}\forall_{x>b_\epsilon}: 1 \geq CDF_{\mathcal{N}(\sqrt{D\mu},\sigma^2/(4\mu))}(\sqrt{D\mu}+x) \geq 1-\epsilon,$$

$$\forall_{\epsilon>0}\exists_{\mathcal{D}>0}\forall_{D>\mathcal{D},D\in\mathbb{N}}\forall_{x>b_\epsilon}: 1 \geq CDF_{\sqrt{\mathcal{N}_+(D\mu,D\sigma^2)}}\left(\sqrt{D\mu}+x\right) > 1-2\epsilon,$$

$$\forall_{\epsilon>0}\exists_{\mathcal{D}>0}\forall_{D>\mathcal{D},D\in\mathbb{N}}\forall_{x>b_\epsilon}:$$
$$\left| CDF_{\mathcal{N}\left(\sqrt{D\mu},\sigma^2/(4\mu)\right)}\left(\sqrt{D\mu}+x\right) - CDF_{\sqrt{\mathcal{N}_+(D\mu,D\sigma^2)}}\left(\sqrt{D\mu}+x\right) \right| < 2\epsilon. \quad (12)$$

Thus,

$$\forall_{\epsilon>0}\exists_{\mathcal{D}>0}\forall_{D>\mathcal{D},D\in\mathbb{N}}\forall_{x\in\mathbb{R}}: \left[ \sqrt{D\mu}-b_\epsilon > 0 \implies \right.$$
$$\left. \left| CDF_{\mathcal{N}\left(\sqrt{D\mu},\sigma^2/(4\mu)\right)}\left(\sqrt{D\mu}+x\right) - CDF_{\sqrt{\mathcal{N}_+(D\mu,D\sigma^2)}}\left(\sqrt{D\mu}+x\right) \right| < 2\epsilon \right], \quad (13)$$

because for any $\epsilon > 0$ we may define $\mathcal{D} := \max\{\mathcal{D}_1,\mathcal{D}_2,\mathcal{D}_3\}$, where $\mathcal{D}_1,\mathcal{D}_2,\mathcal{D}_3$ are taken from (6), (11) and (12).

To simplify,

$$\forall_{\epsilon>0}\exists_{\mathcal{D}>0}\forall_{D>\mathcal{D},D\in\mathbb{N}}\forall_{x\in\mathbb{R}}: \left| CDF_{\mathcal{N}\left(\sqrt{D\mu},\sigma^2/(4\mu)\right)}(x) - CDF_{\sqrt{\mathcal{N}_+(D\mu,D\sigma^2)}}(x) \right| < 2\epsilon, \quad (14)$$

because for any $\epsilon > 0$ we may define $\mathcal{D} := \max\{\mathcal{D}_1, \mathcal{D}_2\}$, where $\mathcal{D}_1, \mathcal{D}_2$ are taken from (4) and (13), making the antecedent true. We also replaced $\sqrt{D\mu} + x$ with $x$, since now the statement holds for all $x \in \mathbb{R}$.

Finally, we combine (3) and (14) using the triangle inequality

$$\forall_{\epsilon>0} \exists_{\mathcal{D}>0} \forall_{D>\mathcal{D}, D\in\mathbb{N}} \forall_{x\in\mathbb{R}} : \left| CDF_{\|\mathbf{Z}\|}(x) - CDF_{\mathcal{N}\left(\sqrt{D\mu}, \sigma^2/(4\mu)\right)}(x) \right| < 3\epsilon, \qquad (15)$$

because for any $\epsilon > 0$ we may define $\mathcal{D} := \max\{\mathcal{D}_1, \mathcal{D}_2\}$, where $\mathcal{D}_1, \mathcal{D}_2$ are taken from (3) and (14), and since it is true for any positive $\epsilon$, we replace $3\epsilon$ with $\epsilon$

$$\forall_{\epsilon>0} \exists_{\mathcal{D}>0} \forall_{D>\mathcal{D}, D\in\mathbb{N}} \forall_{x\in\mathbb{R}} : \left| CDF_{\|\mathbf{Z}\|}(x) - CDF_{\mathcal{N}\left(\sqrt{D\mu}, \sigma^2/(4\mu)\right)}(x) \right| < \epsilon,$$

because for any $\epsilon > 0$ we may define $\mathcal{D} := \mathcal{D}_1$, where $\mathcal{D}_1$ is taken from (15), substituting $\frac{\epsilon}{3}$ for $\epsilon$. □

**Observation 2.2.** *If $\mathbf{Z}$ has a finite mean, and $\overline{\mathbf{Z}}$ is distributed identically to $\mathbf{Z}$, then $\mathbf{Z}$ must be concentrated at a single point.*

*Proof.* Let $\mathbf{Z}, \mathbf{Z}^{(1)}, \mathbf{Z}^{(2)}, \mathbf{Z}^{(3)}, \ldots$ be an infinite sequence of independent and identically distributed random variables. Using induction on $n$ we can show that $\frac{1}{2^n}\left(\mathbf{Z}^{(1)} + \ldots + \mathbf{Z}^{(2^n)}\right)$ is distributed identically to $\mathbf{Z}$. Indeed, for $n = 1$ this is one of the theorem's assumptions. To prove the inductive step let us define

$$\mathbf{A} := \frac{1}{2^n}\left(\mathbf{Z}^{(1)} + \ldots + \mathbf{Z}^{(2^n)}\right),$$

$$\mathbf{B} := \frac{1}{2^n}\left(\mathbf{Z}^{(2^n+1)} + \ldots + \mathbf{Z}^{(2^{n+1})}\right).$$

$\mathbf{A}$ and $\mathbf{B}$ are independent – they are defined as functions of independent variables – and, by the inductive hypothesis, distributed identically to $\mathbf{Z}$. Finally, it is sufficient to observe that

$$\frac{1}{2^{n+1}}\left(\mathbf{Z}^{(1)} + \ldots + \mathbf{Z}^{(2^{n+1})}\right) = \frac{\mathbf{A} + \mathbf{B}}{2}.$$

$\mathbf{Z}$ has finite mean – let us denote it by $\mu$. Let also $\mathbb{N}_+$ be the set of strictly positive natural numbers. By the law of large numbers the sequence $\{\frac{1}{n}(\mathbf{Z}^{(1)} + \ldots + \mathbf{Z}^{(n)})\}_{n\in\mathbb{N}_+}$ converges in probability to $\mu$. The same is true for any infinite subsequence, in particular for $\{\frac{1}{2^n}(\mathbf{Z}^{(1)} + \ldots + \mathbf{Z}^{(2^n)})\}_{n\in\mathbb{N}_+}$, but we have shown that all elements of this subsequence are distributed identically to $\mathbf{Z}$, thus $\mathbf{Z}$ must be concentrated at $\mu$.

□

**Observation 3.1.** *If $\mathbf{Z}$ has uniform distribution on the zero-centred sphere of radius $R > 0$, then $f^{SL}$ does not change the distribution of $\mathbf{Z}$.*

*Proof.* Let $\mathbf{Z}, \mathbf{Z}^{(1)}, \mathbf{Z}^{(2)}$ be independent and identically distributed. Let $\lambda \in [0, 1]$ be a fixed real number. The random variable $f^{SL}_{\mathbf{Z}^{(1)}, \mathbf{Z}^{(2)}}(\lambda)$ is defined almost everywhere (with the exception of parallel samples from $\mathbf{Z}^{(1)}, \mathbf{Z}^{(2)}$) and is also concentrated on the zero-centred sphere of radius $R$ (because if $\|x_1\| = \|x_2\|$, then $\|f^{SL}_{x_1,x_2}(\lambda)\| = \|x_1\| = \|x_2\|$).

Let $iso$ be any linear isometry of the latent space. $\|iso(x)\| = \|x\|$, thus $iso$ is also an isometry of the zero-centred sphere of radius $R$. Additionally, we have

$$iso\left(f^{SL}_{x_1,x_2}(\lambda)\right) = iso\left(\frac{\sin\left[(1-\lambda)\Omega\right]}{\sin\Omega}x_1 + \frac{\sin[\lambda\Omega]}{\sin\Omega}x_2\right)$$

$$= \frac{\sin\left[(1-\lambda)\Omega\right]}{\sin\Omega}iso(x_1) + \frac{\sin[\lambda\Omega]}{\sin\Omega}iso(x_2)$$

$$= f^{SL}_{iso(x_1),iso(x_2)}(\lambda)$$

and the last equality holds because the isometry does not change the angle $\Omega$ between $x_1$ and $x_2$.

Thus, $iso\big(f^{SL}_{\mathbf{Z}^{(1)},\mathbf{Z}^{(2)}}(\lambda)\big) = f^{SL}_{iso(\mathbf{Z}^{(1)}),iso(\mathbf{Z}^{(2)})}(\lambda)$, and this is distributed identically to $f^{SL}_{\mathbf{Z}^{(1)},\mathbf{Z}^{(2)}}(\lambda)$, because $\mathbf{Z}^{(1)}, \mathbf{Z}^{(2)}$, both uniform distributions, are invariant to $iso$.

In that case, $f^{SL}_{\mathbf{Z}^{(1)},\mathbf{Z}^{(2)}}(\lambda)$ is concentrated on the zero-centred sphere of radius $R$ and invariant to all linear isometries of the latent space. The only distribution having these properties is the uniform distribution on the sphere.

$\square$

**Observation 3.2.** *If $\mathbf{Z} \sim \mathcal{N}(\mathbf{0}, \mathbf{I})$, then $f^N$ does not change the distribution of $\mathbf{Z}$.*

*Proof.* Let $\mathbf{Z}, \mathbf{Z}^{(1)}, \mathbf{Z}^{(2)}$ be independent and identically distributed. Let $\lambda \in [0,1]$ be a fixed real number. The random variables $\mathbf{Z}^{(1)}$ and $\mathbf{Z}^{(2)}$ are both distributed according to $\mathcal{N}(\mathbf{0}, \mathbf{I})$. Using the definition of $f^N$ and elementary properties of the normal distribution we conclude

$$f^N_{\mathbf{Z}^{(1)},\mathbf{Z}^{(2)}}(\lambda) = \frac{(1-\lambda)\mathbf{Z}^{(1)} + \lambda\mathbf{Z}^{(2)}}{\sqrt{(1-\lambda)^2 + \lambda^2}} \sim \mathcal{N}\Big(\frac{(1-\lambda)\mathbf{0} + \lambda\mathbf{0}}{\sqrt{(1-\lambda)^2 + \lambda^2}}, \frac{(1-\lambda)^2\mathbf{I} + \lambda^2\mathbf{I}}{(1-\lambda)^2 + \lambda^2}\Big) = \mathcal{N}(\mathbf{0}, \mathbf{I}).$$

$\square$

**Observation 3.3.** *With the above assumptions about $g$ the Cauchy-linear interpolation does not change the distribution of $\mathbf{Z}$.*

*Proof.* Let $\mathbf{Z}, \mathbf{Z}^{(1)}, \mathbf{Z}^{(2)}$ be independent and identically distributed. Let $\lambda \in [0,1]$ be a fixed real number. First observe that $g^{-1}(\mathbf{Z}^{(1)})$ and $g^{-1}(\mathbf{Z}^{(2)})$ are independent (because $\mathbf{Z}^{(1)}, \mathbf{Z}^{(2)}$ are independent) and distributed identically to $\mathbf{C}$ (property of $g$). Likewise, $(1-\lambda)g^{-1}(\mathbf{Z}^{(1)}) + \lambda g^{-1}(\mathbf{Z}^{(2)}) \sim \mathbf{C}$ (property of the Cauchy distribution). Therefore, $g\big((1-\lambda)g^{-1}(\mathbf{Z}^{(1)}) + \lambda g^{-1}(\mathbf{Z}^{(1)})\big) \sim \mathbf{Z}$ (property of $g$). $\square$

**Observation 3.4.** *With the above assumptions about $g$, the spherical Cauchy-linear interpolation does not change the distribution of $\mathbf{Z}$ if the $\mathbf{Z}$ distribution is isotropic.*

*Proof.* Let $\mathbf{Z}, \mathbf{Z}^{(1)}, \mathbf{Z}^{(2)}$ be independent and identically distributed. Let $\lambda \in [0,1]$ be a fixed real number. The following statements are straightforward consequences of $\mathbf{Z}^{(1)}, \mathbf{Z}^{(2)}$ being isotropic (and also independent).

1. The random variables $\dfrac{\mathbf{Z}^{(1)}}{\|\mathbf{Z}^{(1)}\|}, \dfrac{\mathbf{Z}^{(2)}}{\|\mathbf{Z}^{(2)}\|}, \|\mathbf{Z}^{(1)}\|, \|\mathbf{Z}^{(2)}\|$ are independent,

2. $\|\mathbf{Z}^{(1)}\|$ and $\|\mathbf{Z}^{(2)}\|$ are both distributed identically to $\|\mathbf{Z}\|$,

3. $\dfrac{\mathbf{Z}^{(1)}}{\|\mathbf{Z}^{(1)}\|}$ and $\dfrac{\mathbf{Z}^{(2)}}{\|\mathbf{Z}^{(2)}\|}$ are both distributed uniformly on the sphere of radius 1.

The next two statements are consequences of Observations 3.1 and 3.3 respectively.

4. The random variable $\dfrac{f^{SCL}_{\mathbf{Z}^{(1)},\mathbf{Z}^{(2)}}(\lambda)}{\|f^{SCL}_{\mathbf{Z}^{(1)},\mathbf{Z}^{(2)}}(\lambda)\|} = \dfrac{\sin\left[(1-\lambda)\Omega\right]}{\sin\Omega}\dfrac{\mathbf{Z}^{(1)}}{\|\mathbf{Z}^{(1)}\|} + \dfrac{\sin[\lambda\Omega]}{\sin\Omega}\dfrac{\mathbf{Z}^{(2)}}{\|\mathbf{Z}^{(2)}\|}$ is distributed uniformly on the unit sphere.

5. The random variable $\|f^{SCL}_{\mathbf{Z}^{(1)},\mathbf{Z}^{(2)}}(\lambda)\| = g\big((1-\lambda)g^{-1}(\|\mathbf{Z}^{(1)}\|) + \lambda g^{-1}(\|\mathbf{Z}^{(2)}\|)\big)$ is distributed identically to $\|\mathbf{Z}\|$.

$\dfrac{f^{SCL}_{\mathbf{Z}^{(1)},\mathbf{Z}^{(2)}}(\lambda)}{\|f^{SCL}_{\mathbf{Z}^{(1)},\mathbf{Z}^{(2)}}(\lambda)\|}$ and $\|f^{SCL}_{\mathbf{Z}^{(1)},\mathbf{Z}^{(2)}}(\lambda)\|$ are independent, because they are functions of independent random variables ($\Omega$ is a function of $\dfrac{\mathbf{Z}^{(1)}}{\|\mathbf{Z}^{(1)}\|}$ and $\dfrac{\mathbf{Z}^{(2)}}{\|\mathbf{Z}^{(2)}\|}$), therefore $f^{SCL}_{\mathbf{Z}^{(1)},\mathbf{Z}^{(2)}}(\lambda)$ is isotropic. Using

the statement 5. and the fact that two isotropic probability distributions are equal if and only if the distributions of their euclidean norms are equal we conclude that $f_{\mathbf{Z}^{(1)},\mathbf{Z}^{(2)}}^{SCL}(\lambda)$ is distributed identically to $\mathbf{Z}$. $\square$

## C THE CAUCHY DISTRIBUTION – SAMPLES AND INTERPOLATIONS

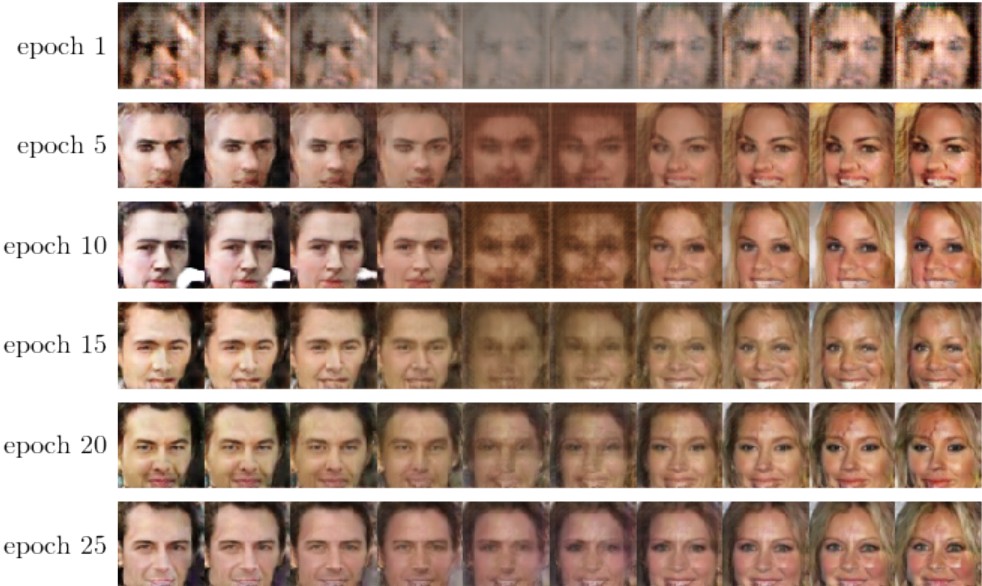

Figure 6: Emergence of sensible samples decoded near the origin of the latent space throughout the training process. Demonstrated using interpolations between opposite vectors sampled from the latent space.

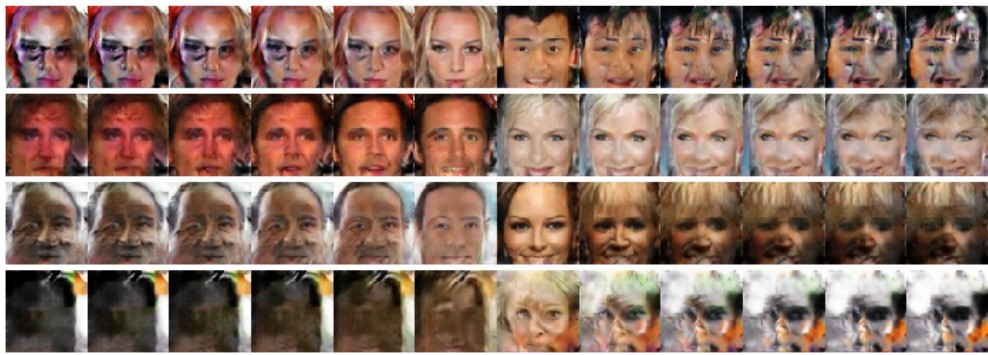

Figure 7: Linear interpolations between hand-picked points from tails of the Cauchy distribution.

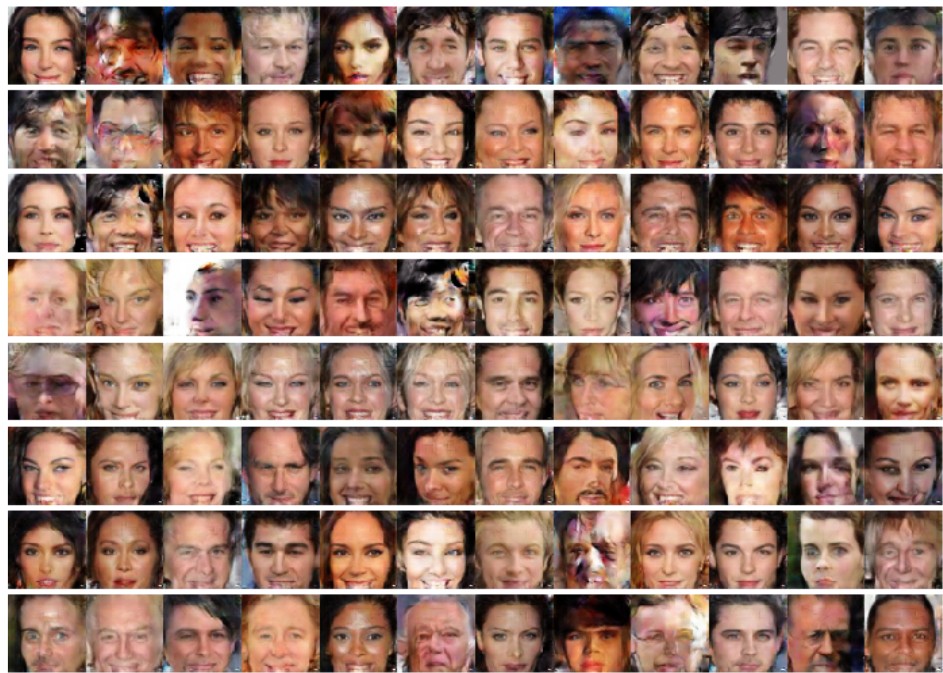

Figure 8: Generated images from samples from the Cauchy distribution, with occasional "failed" images from tails of the distribution.

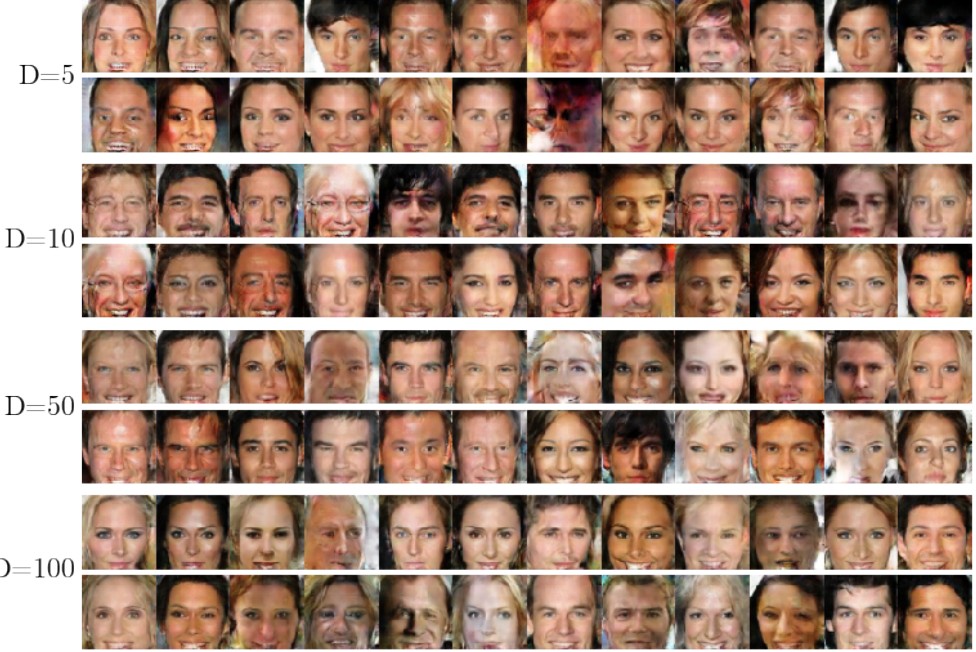

Figure 9: Images generated from samples from the Cauchy distribution, with varying dimension of the latent space.

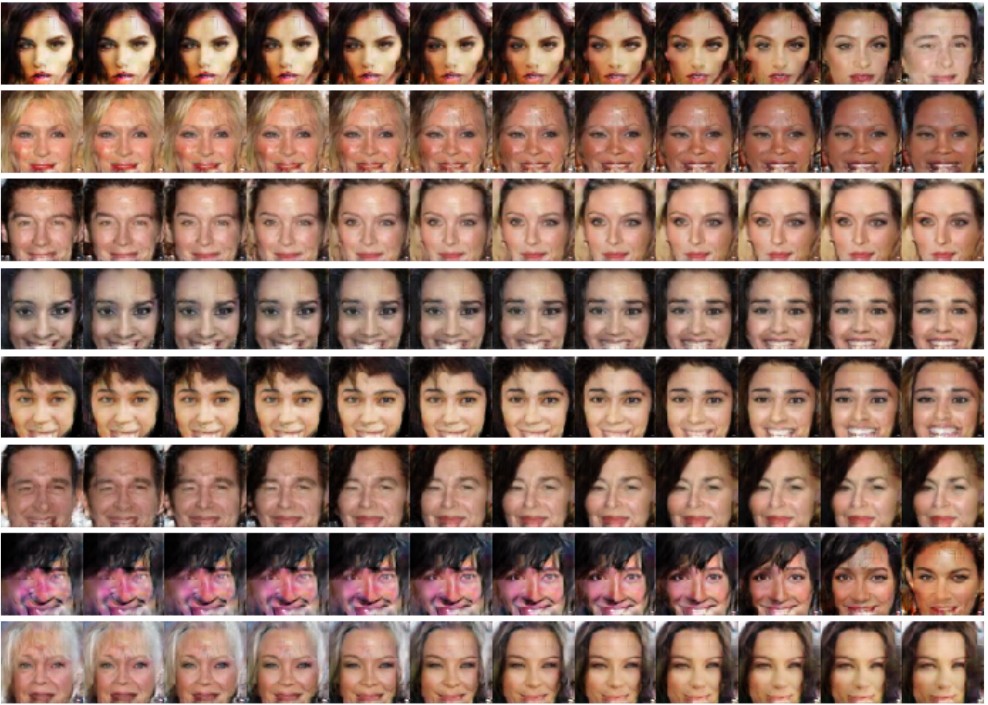

Figure 10: Linear interpolations between random latent vectors. The model was trained using the Cauchy distribution.

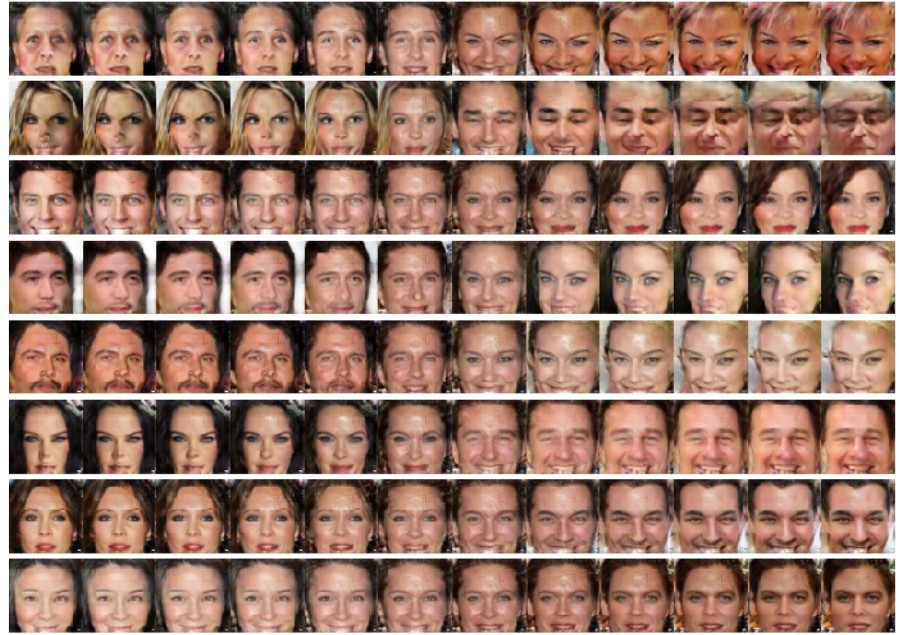

Figure 11: Linear interpolations between opposite latent vectors. The model was trained using the Cauchy distribution.

# D  MORE CAUCHY-LINEAR AND SPHERICAL CAUCHY-LINEAR INTERPOLATIONS

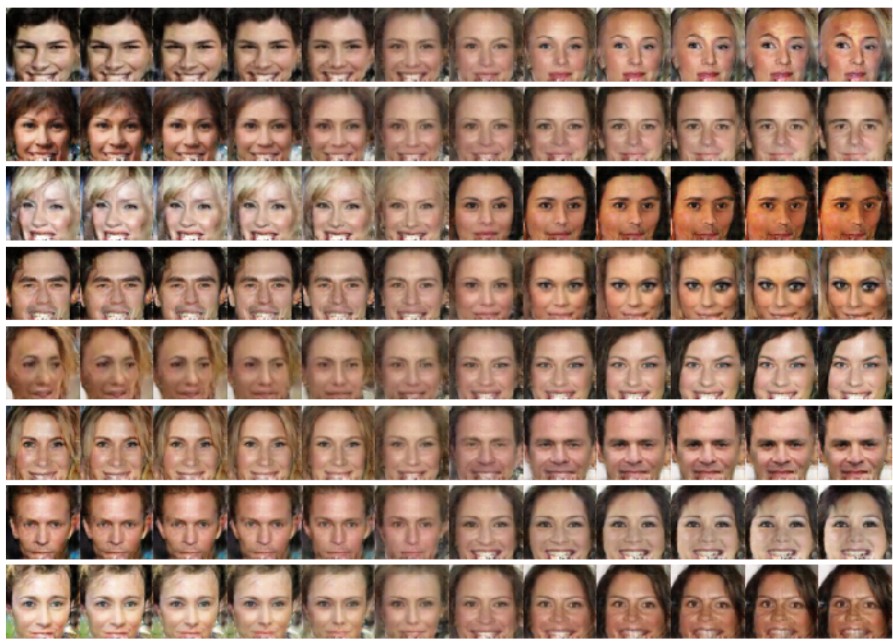

Figure 12: Cauchy-linear interpolations between opposite latent vectors. The model was trained using the normal distribution.

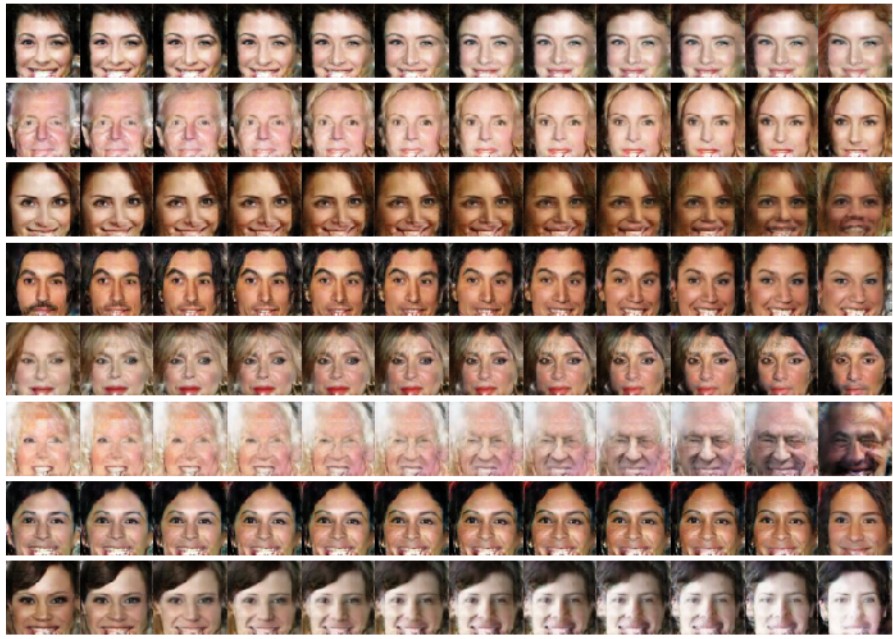

Figure 13: Cauchy-linear interpolations between random latent vectors. The model was trained using the normal distribution.

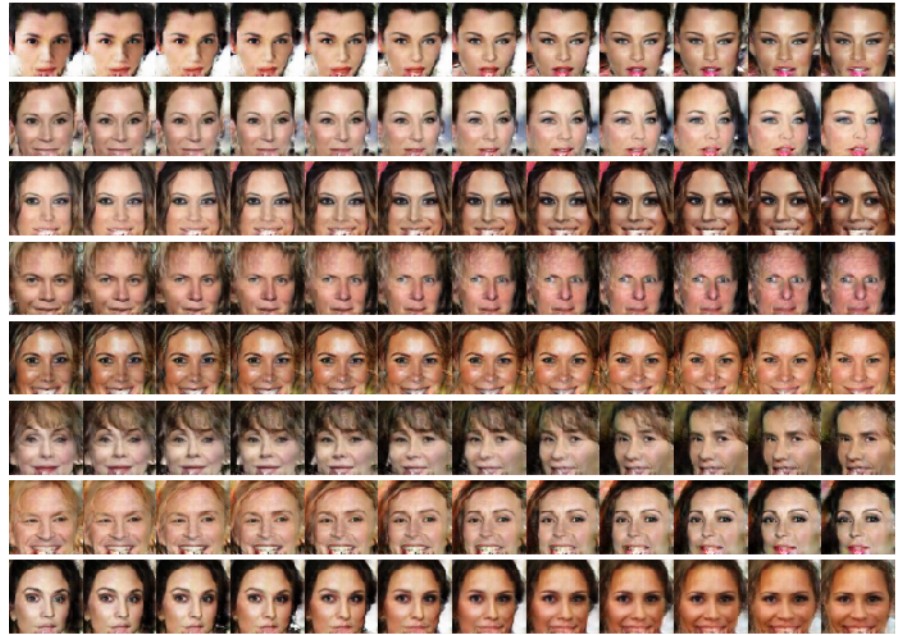

Figure 14: Spherical Cauchy-linear interpolations between random latent vectors. The model was trained using the normal distribution.

