# OpenReview forum: "Distribution-Interpolation Trade off in Generative Models"
_ICLR.cc/2019/Conference_

### Official Review · AnonReviewer1 · 2018-10-29
**An interesting analysis of linear interpolations with respect to the prior in the latent space, however the paper needs some improvements especially in the motivation and its implications.**

**Rating:** 5
**Confidence:** 4

**Review:**

The authors study the problem of when the linear interpolant between two random variables follows the same distribution. This is related to the prior distribution of an implicit generative model. In the paper, the authors show that the Cauchy distribution has such a property, however due to the heavy-tails is not particularly useful. In addition, they propose a non-linear interpolation that naturally has this property.

Technically the paper in my opinion is solid. Also, the paper is ok-written, but I think it needs improvements (see comments).

Comments:

#1) In my opinion the motivation is not very clear and should be improved. In the paper is mentioned that the goal of shortest path interpolation is to get smooth transformations. So, in principle, I am really skeptical when the linear interpolant is utilized as the shortest path. Even then, what is the actual benefit of having the property that the linear interpolants follow the same distribution as the prior? How this is related to smoother transformations? What I understand is that, if we interpolate between several random samples, we will get less samples near the origin, and additionally, these samples will follow the prior? But how this induces smoothness in the overall transformation? I think this should be explained properly in the text i.e. why is it interesting to solve the proposed problem.

#2) From Observation 2.2. we should realize that the distribution matching property holds if the distribution has infinite mean? I think that this is implicitly mentioned in Section 2.2. paragraph 1, but I believe that it should be explicitly stated.

#3) Fig.1 does not show something interesting, and if it does it is not explained. In Fig. 2 I think that interpolations between the same images should be provided such that to have a direct comparison. Also, in Fig. 3 the norm of Z can be shown in order to be clear that the Cauchy distribution has the desired property.

#4) Section 2.2. paragraph 6, first sentence. Here it is stated that the distribution "must be trivial or heavy-tailed". This refers only to the Cauchy distribution? Since earlier the condition was the infinite mean. How these are related? Needs clarification in the text.

#4) In Figure 4, I believe that the norms of the interpolants should be presented as well, such that to show if the desired property is true. Also in Figure 5, what we should see? What are the improvements when using the proposed non-linear interpolation?


Minor comments:

#1) Section 1.2. paragraph 2. For each trained model the latent space usually has different structure e.g. different untrained regions. So I believe that interpolations is not the proper way to compare different models.

#2) Section 1.3 paragraph 1, in my opinion the term "pathological" should be explained precisely here. So it makes clear to the reader what he should expect.

#3) Section 2.2. paragraph 2. The coordinate-wise implies that some Z_i are near zero and some others significantly larger?

In generally, I like the presented analysis. However, I do not fully understand the motivation. I think that choosing the shortest path guarantees smooth transformations. I do not see why the distribution matching property provides smoother transformations. To my understanding, this is simply a way to generate less samples near the origin, but this does not directly means smoother transformations of the generated images. I believe that the motivation and the actual implications of the discussed property have to be explained better.

---

> ### Author Response · Authors · 2018-11-13
> **Response to the review**
>
> Firstly, we are grateful for the insights provided by the reviewer. We will address the issues in the order they were given.
>
> Comment #1)
>
> It is now apparent to us that our motivation is unclear and needs to be worked on as this was brought up by all three reviewers.
>
> We argue that using interpolations that do not cause the distribution mismatch has a one, big advantage over linear interpolations: they are compatible with the model’s objective. We do not argue that this automatically yields smoother transitions, although we point to this as one of the reasons that the linear interpolations may be less smooth.
>
> We argue that the interpolations may be used twofold: to analyse the model performance or to generate smooth transitions between two selected endpoints. In the first case we must use an interpolation that does not cause the distribution mismatch - it is completely unjustified to train a model on one probability distribution, and then evaluate on another. The second goal is possible to achieve even if the model failed to converge on the latent probability distribution - we mention the related work in the last paragraph of section 1.1.
>
> In this work we focus solely on the first case.
>
> The main issue with the linear interpolation is that it cannot be used in neither of the cases, as it causes the distribution mismatch and does not take the learned latent manifold into consideration. On the other hand, in the case of models that do work well with linear interpolations it is still an interesting question as to why they manage to extend the latent manifold to include points seemingly out of prior distribution. Thus we do not advise to completely abandon linear interpolations, but simply to use them in a proper context.
>
> Comment #2)
>
> The infinite mean is the necessary condition for the distribution matching property, but it is not sufficient. We use the Cauchy distribution (and its variants) as it is the only probability distribution known to us that has the matching property.
>
> Comment #3)
>
> Fig.1 is supposed to show that models with Cauchy distribution and other proposed distribution from the literature yield similar samples. We agree that this may not convey enough information and removed it from the final version.
>
> With fig. 2 we agree that a comparison between the same endpoints would be more comprehensive. We tried to show such comparisons where it was easily achievable (e.g. fig 6), but in the case of fig. 2 the samples come from different distributions with broadly different norms. This results in a need for either an additional encoder or an optimisation scheme to find the same/similar image in those two models. We will try to perform such experiments and change the figure.
>
> As for fig. 3 we need to be sure that we understand the issue correctly. Would the reviewer like to see a distribution of interpolation midpoints for Cauchy distribution? Fig. 3 does not concern with interpolations, and its purpose is to visualise the conclusion of the observation 2.1, we will clarify this in the caption of the figure.
> It is possible to add such a comparison for norm distribution of interpolation midpoints later on in the text.
>
> Comment #4)
>
> We agree that this needs a clarification in the text. One of the consequence of a distribution having undefined expected value is being heavy-tailed. If a distribution is not heavy-tailed, then its tails are bounded by the exponential distribution, which in turn means that it has finite mean.
> We mention heavy tails as they may have a direct impact on the training of deep models, which prefer normalized input.
>
> Comment #5)
>
> For fig. 4 we need some clarification, what does the reviewer mean by “the norms”? Would labelling axes allow comparison between the norms the sub figures?
>
> Fig. 5 does not show any of the interpolations as improved over other, its purpose is to visualize the interpolations on the same pair of points, perhaps the reviewer implies that it is redundant to the fig. 4?
>
> Minor Comment #1)
>
> To clarify, we used the same architecture and hyperparameters with the same objective. We looked into the claim that samples near the origin of the latent space result in flawed images. With that in mind we used interpolations that traverse that region near the origin as a way of comparing the results.
>
> Minor Comment #2)
>
> We agree that this needs an explanation, we will clarify that we mean a distribution that does not have a finite mean.
>
> Minor Comment #3)
>
> Yes, exactly that.
>
>
> To conclude, we have heard from all three reviewers that the paper needs a better motivation and will improve it in the final version as well as clarify the other mentioned issues.

---

> > ### Comment · AnonReviewer1 · 2018-11-23
> > **Response to authors**
> >
> > - Based on the other reviews and the authors rebuttal, I think all of us agree that the analysis in the paper is solid, but the clarity of the paper is lacking. For instance I am not sure if the paper proposes a way to compare generative models or is a paper that studies and presents a solution to a more general problem i.e. distribution matching property.
> >
> > - In general, I like a lot the mathematical analysis, but I feel that the paper should be improved. The content is good, but in my opinion a bit poorly presented in the current form.
> >
> > - I appreciate that the reviewers answered all the questions. Considering the Comment #5), I would like to see in the current Fig. 2 the corresponding result similar to the current Fig. 4.

---

> > > ### Author Response · Authors · 2018-11-23
> > > **Re: Response to authors**
> > >
> > > We tried to clarify our motivation and contribution in the updated version. We aimed to emphasise the main point of our paper, the distribution mismatch, and its negative influence on the evaluation process. We do not propose any new evaluation methods, but  formally prove that any analysis with an interpolation should incorporate such an algorithm, that does not cause the distribution mismatch. In the paper’s conclusions we have added a short paragraph listing which interpolations we postulate to be used in which case.
> > >
> > > > Considering the Comment #5), I would like to see in the current Fig. 2 the corresponding result similar to the current Fig. 4.
> > >
> > > We attempted to add additional distributions for the midpoints to Fig. 2, but it made the graphs cluttered to a degree of being almost unreadable. If we understand the reviewer’s intentions correctly, such a plot would resemble current Fig. 4 without the Cauchy interpolation histogram (in orange).

---

### Official Review · AnonReviewer3 · 2018-10-31
**Interesting ideas and observations;  very early work**

**Rating:** 7
**Confidence:** 3

**Review:**

== Paper overview ==
Given a latent variable model (deep generative model), the paper ask how we should interpolate in the latent space. The key idea is to derive a natural interpolant from the prior distribution p(z), where z is the latent variable. The idea is that the interpolation function you apply to a variable z should not change the distribution of z of the start and end points of the interpolation curve are identically distribution. Example: consider two unit-length points drawn from a standard Gaussian, then linear interpolation of these points will result in points of smaller norm and hence different distribution. Differerent priors and corresponding interpolants are demonstrated and discussed. Empirical results are more of an illustrative nature.

== Pros/cons ==
+ The paper contribute a new and relevant point to an ongoing discussion on the geometry of the latent space.
+ The key point is well-articulated and relevant mathematical details are derived in detail along the way.

- I have some concerns about the idea itself (see below); yet, while I disagree with some of the presented view points, I don't think that diminishes the contribution.
- The empirical evaluation hardly qualifies as such. A few image interpolations are shown, but it is unclear what conclusions can really be drawn from this. In the end, it remains unclear to me which approach to interpolation is better.

== Concerns / debate ==
I have some concerns about the key idea of the paper (in essence, I find it overly simplistic), but I nonetheless find that the paper brings an interesting new idea to the table.

1) In section 1.1, the authors state "one would expect the latent space to be organized in a way that reflects the internal structure of the training dataset". My simple counter-question is: why? I know that this is common intuition, but I don't see anything in the cost functions of e.g. VAEs or GANs to make the statement true. Generative models, as far as I can see, only assume that the latent variables are somehow compressed versions of the data points; no assumptions on structure seems to be made.

2) Later in the same section, the authors state "In absence of any additional knowledge about the latent space, it feels natural to use the Euclidean metric". Same question: why? Again, I know that this is a common assumption, but, again, there is nothing in the models that seem to actually justify such an assumption. I agree that it would be nice to have a Euclidean latent space, but doesn't make it so.

3) In practice, we often see "holes" in the "cloud" of latent variables, that is regions of the latent space where only little data resides. I would argue that a good interpolant should not cross over a hole in the data manifold; none of the presented interpolants can satisfy this as they only depend no the start and end points, but not on the actual distribution of the latent points. So if the data does not fit the prior or are not iid, then the proposed interpolants will most likely perform poorly. A recent arXiv paper discuss one way to deal with such holes: https://arxiv.org/abs/1806.04994

---

> ### Author Response · Authors · 2018-11-13
> **Response to the review**
>
> We would like to thank the reviewer for valuable comments. We will address the concerns below.
>
> Concerns 1) and 2)
>
> It might have been unclear, but our intention of saying “one would expect the latent space to be organised in a way that reflects the internal structure of the learning set” and “in absence of any additional knowledge about the latent space, it feels natural to use the Euclidean metric” was not to say that the training objective assures that the model has those properties, but it would be welcome to additionally achieve this goal.
>
> The purpose of this chapter is to justify the use of interpolations in analysis of trained models - as a tool to check the aforementioned properties. We also try to explain the prevalence of linear interpolation in the literature by introducing the concept of shortest path, which is, in our minds, one of its two main reasons (the second one being simplicity).
>
> We shall extend the motivation of our paper to better explain this objectives.
>
> Concern 3)
>
> In our understanding, there are two main ways of analysing a trained model using interpolations.
>
> The first one is using interpolations that correspond to the training objective, and checking if the decoded samples are viable. If not, then we may conclude that the model has failed.
>
> The second one assumes that the model has learned a manifold that does not correspond to the latent prior, thus we need to define more complex interpolations to capture the manifold’s structure. Then, instead of evaluating the performance on the prior distribution, we may analyse the interpolation itself.
>
> As we have noted in the last paragraph of subsection 1.1, the two approaches are complementary and equally important, but in this paper we focus on the first one. We criticise the use of linear interpolations, as they result in a distribution mismatch. Our main objective was to study this problem and introduce interpolations which are theoretically correct - then we might be certain that the potential failure to generate proper samples was caused by a faulty model and not the distribution mismatch between given latent points and the prior distribution.
>
> We agree that the second approach might be considered better, because, if done correctly, it also allows us to generate smooth interpolations even if the model has failed to converge on the given latent prior. On the other hand, if we are mainly concerned with evaluating the model, it might be safer to use the simpler approach. We shall emphasise this fact in the last paragraph of section 1.1.
>
>
> We will clarify the wording in paragraphs from concerns 1) and 2) and update the motivation to better reflect our intent. We will also add additional conclusion stating that there is no “best” interpolation scheme, as the ones mentioned in the paper serve different purposes.

---

> > ### Comment · AnonReviewer3 · 2018-11-23
> > **Has the paper been updated?**
> >
> > Thanks for the replies. I think it is clear from all reviews (and the authors seem to agree), that the paper needs motivational work. Given that ICLR actually supports revisions of the paper, then I am willing to re-read an updated version of the paper. Has the mentioned updated to the motivation already been incorporated into the paper?
> >
> > Another remark: there seems to be some debate about the "smoothness" of different interpolation functions. I think something is missing from that debate: I would argue that the smoothness of a given interpolation is not so much dependent on the path along with you interpolate, but rather on the speed at which you traverse this path. I guess the mechanisms of this paper can only inform us about the shape of an interpolation path, but not about the speed with which it should be traversed -- is that correct?

---

> > > ### Author Response · Authors · 2018-11-23
> > > **Yes, we have updated the paper.**
> > >
> > > Yes, we have updated the paper with our best intentions to clarify our motivation and contributions. With that in mind the bulk of the changes were made in the introduction and conclusions of the text. Furthermore, a couple of figures were changed or updated to incorporate the feedback from other reviewers.
> > >
> > > In the case of our interpolation methods, the speed at which they are traversed is directly enforced by the distribution. This can be observed in the Fig. 3, where the dots on the lines represent subsequent interpolation points. If one were to change that speed, e.g. to constant (as in the linear interpolation), this would cause the distribution mismatch again. In conclusion, our method informs about both the path and the speed on the interpolation.

---

### Official Review · AnonReviewer2 · 2018-10-31
**This paper discusses an interesting topic without any clear conclusions.**

**Rating:** 6
**Confidence:** 4

**Review:**

The paper discusses linear interpolations in the latent space, which is one of the common ways used nowadays to evaluate a  quality of implicit generative models. More precisely, what researchers often do in this field is to (a) take a trained model (which often comes with a "decoder" or a "generator", that is a function mapping a noise sampled from a prior distribution Pz defined over the latent space Z to the data space), (b) sample two independent points Z1 and Z2 from Pz, and (c) report images obtained by decoding linear interpolations between Z1 and Z2 in the latent space. Researchers often tend to judge the quality of the model based on these interpolations, concluding that the model performs poorly if the interpolations don't look realistic and vice versa. The authors of the paper argue that this procedure has drawbacks, because in typical modern use cases (Gaussian / uniform prior Pz) the aforementioned interpolations are not distributed according to Pz anymore, and thus are likely to be out of the domain where decoder was actually trained.

I would say the main contributions of the paper are:
(1) The sole fact that the paper highlights the problems of linear interpolation based evaluation is already important.
(2) Observation 2.2, stating that if (a) Pz has a finite mean and (b) aforementioned linear interpolations are still distributed according to Pz, then Pz is a Dirac distribution (a point mass).
(3) The authors notice that Cauchy distribution satisfies point (a) from (2), but as a result does not have a mean. The authors present some set of experiments, where DCGAN generator is trained on the CelebA dataset with the Cauchy prior. The interpolations supposedly look nice but the sampling gets problematic, because a heavy tailed Cauchy often results in the Z samples with excessively large norm, where generator performs poorly.
(4) The authors propose several non-linear ways to interpolate, which keep the prior distribution unchanged (Sections 3.4 and 3.5). In other words, instead of using a linear interpolation and Pz compatible with it (which is necessarily is heavy tailed as shown in Observation 2.2), the authors propose to use non linear interpolations which work with nicer priors Pz, in particular the ones with finite mean.

I think this topic is very interesting and important, given there is still an unfortunate lack of well-behaved and widely accepted evaluation metrics in the field of unsupervised generative models.

Unfortunately, I felt the exposition of the paper was rather confusing and, more importantly, I did not find a clear goal of the paper or any concrete conclusions. One possible conclusion could be that the generative modelling community should stop reporting the linear interpolations. However, I feel the paper is lacking a convincing evidence (from what I could find the authors base all the conclusions on one set of similar experiments performed with one generator architecture on one data set) in order to be viewed as a significant contribution to the generative modeling field. On the other hand, the paper has not enough insights to constitute a significant theoretical contribution (I would expect Observation 2.2 to be already known in the probability field).

Overall, I have to conclude that the paper is not ready to be published but I am willing to give it a chance.

---

> ### Author Response · Authors · 2018-11-13
> **Response to the review**
>
> First of all, we would like to thank the reviewer for theirs insights. We agree that the exposition of the paper could be written more clearly and we will try to improve this. We will address the main issues here.
>
> Our main conclusion in the paper states that we recommend to use our interpolations to analyse the generative models in conjunction with linear interpolations. What we mean by that is that our interpolations should be used when analysing whether a model has learned its objective. Linear interpolations, on the other hand, are useful if one would like to evaluate generation on out-of-distribution points.
>
> As an example let us imagine that we have a trained generative model and would like to analyse it using interpolations. If samples from linear interpolations are sensible, then we usually conclude that the model performs well (or even better than expected, but more on that later). However if the linear interpolations yield flawed samples, there might be at least two different reasons. The generative model might be poorly trained, or the model is well trained in regards to its objective and the reason for flawed inference is the distribution mismatch. To isolate the first case we need to use interpolations that do not cause the mismatch, e.g. the ones we propose in this work. Certainly, we cannot use linear interpolations.
>
> We would like to note that there may be similar cases when one needs to be careful when deriving conclusions from results of linear interpolations, as any faults that may arise are possibly due to the distribution mismatch and not model misspecification.
>
> We do not propose to abandon linear interpolations at all, as we believe it is a fairly interesting question as to why the model yields proper samples on the linear interpolations despite the distribution mismatch. We mention this in our conclusion.
>
> We base our conclusions on strictly theoretical arguments. Additionally, we provide a formal proof that our approach does not suffer from the distribution mismatch. We strongly agree with the reviewer that there is no consensus on an empirical method to evaluate interpolation quality, thus any empirical evaluation would be purely visual. Hence, we use the DCGAN only as an illustration of the problem.
>
> As for the more practical contributions, we provide a general scheme of creating distribution-matching interpolations. We argue that the idea of transforming the distribution to the Cauchy distribution and only then using linear interpolations is easily applicable to a broad set of problems.
>
> We agree that the observation 2.2 is one of the most basic parts of our work, as it is a straightforward consequence of the Law of Large Numbers. Keeping this is mind, we find it even more surprising that this fact wasn’t brought up in the field, to the point that one can find works that, correctly, pinpoint the interpolation distribution mismatch as a serious issue that needs to be tackled. Using linear interpolations to validate if the model has learned its objective is a serious theoretical flaw. We show how to avoid this issue, providing two separate solutions: using a specific latent prior or a distribution matching interpolation. We also prove that it is impossible to have both linear interpolation and distribution match - hence the titular trade off.
>
>
> We hope that this clears most of reviewer’s concern, we will update the text accordingly with an emphasis on our motivation.

---

> > ### Comment · AnonReviewer2 · 2018-11-29
> > **Willing to give it a chance.**
> >
> > I thank the authors for their response.
> >
> > I still think that the paper touches on important questions and provides some novel insights. I am willing to raise my score but not going to strongly fight for the paper. I think the paper can make a much larger impact if the authors take more time to perform a more systematic study (both theoretical and empirical) making their message solid. In the current form, the paper has a good chance of not being noticed which would be a shame.

---

> > > ### Author Response · Authors · 2018-11-29
> > > **Re: Willing to give it a chance.**
> > >
> > > We thank the reviewer for the comments and assure that we will do our best to highlight the issues presented in our work to the community.

---

### Meta-Review · Area_Chair1 · 2018-12-17

**Confidence:** 5
**Recommendation:** Accept (Poster)

**Metareview:**

All the reviewers and AC agrees that the main strength of the paper that it studies a rather important question of the validity of using linear interpolation in evaluating GANs. The paper gives concrete examples and theoretical and empirical analysis that shows linear interpolation is not a great idea. The potential weakness is that the paper doesn't provide a very convincing new evaluation to replace the linear interpolation. However, given that it's largely unclear what are the right evaluations for GANs, the AC thinks the "negative result" about linear interpolation already deserves an ICLR paper.